# Human Health Risk Assessment from Lead Exposure through Consumption of Raw Cow Milk from Free-Range Cattle Reared in the Vicinity of a Lead–Zinc Mine in Kabwe

**DOI:** 10.3390/ijerph19084757

**Published:** 2022-04-14

**Authors:** Golden Zyambo, John Yabe, Kaampwe Muzandu, Ethel M’kandawire, Kennedy Choongo, Andrew Kataba, Kenneth Chawinga, Allan Liazambi, Shouta M. M Nakayama, Hokuto Nakata, Mayumi Ishizuka

**Affiliations:** 1School of Veterinary Medicine, The University of Zambia, P.O. Box 32379, Lusaka 10101, Zambia; goldzgambo@gmail.com (G.Z.); kmuzandu@yahoo.com (K.M.); ethel.mkandawire@unza.zm (E.M.); kennedychoongo@yahoo.ie (K.C.); andrewkataba@gmail.com (A.K.); shoutanakayama0219@gmail.com (S.M.M.N.); 2Laboratory of Toxicology, Department of Environmental Veterinary Sciences, Faculty of Veterinary Medicine, Hokkaido University, Kita 18 Nishi 9, Kita-ku, Sapporo 060-0818, Japan; hokuto.nakata@vetmed.hokudai.ac.jp; 3School of Veterinary Medicine, University of Namibia, P.O. Box 13301, Windhoek 10005, Namibia; 4Central Province Veterinary Office, 53 Pauling Street, Kabwe P.O. Box 80285, Zambia; dr.kchawinga@gmail.com (K.C.); aliazambi@gmail.com (A.L.)

**Keywords:** human health risk, milk, lead, food safety, ingestion

## Abstract

Lead (Pb) contamination in the environment affects both humans and animals. Chronic exposure to Pb via dietary intake of animal products such as milk from contaminated areas poses a health risk to consumers; therefore, the present study investigated Pb contamination in cow milk and its health risk impact on humans through consumption of milk from cattle reared in the proximity of a Pb–Zn mine in Kabwe, Zambia. Fresh milk samples were collected from cows from Kang’omba (KN), Kafulamse (KF), Mpima (MP), Mukobeko (MK), and Munga (MN) farming areas. Pb determination was performed using Graphite Flame Absorption Atomic Spectrophotometry (GFAAS). Cow milk Pb levels showed different concentration patterns according to season, distance, and location of the farms from the Pb–Zn mine. The overall mean Pb levels were ranged 0.60–2.22 µg/kg and 0.50–4.24 µg/kg in the wet and dry seasons, respectively. The mean Pb concentration, chronic daily intake (CDIs), target hazard quotients (THQs), and incremental lifetime cancer risk (ILCR) results obtained were all within the permissible limits of 20 µg/kg, 3 and 12.5 µg/kg-BW/day, <1 and 10^−4^ to10^−6^, respectively. In conclusion, although Pb was detected in milk from cows reared in Kabwe, the health risk effects of Pb exposure associated with the consumption of milk in both adults and children were negligible.

## 1. Introduction

Lead (Pb) is a toxic and possible carcinogenic element [1] that results from multiple sources in the environment [2]. Because of its widespread use, Pb has caused extensive environmental contamination and health problems in many parts of the world [3]. Irrespective of the source, human exposure to Pb is one of the most serious health problems facing populations, especially children [4]. In animals, grazing on contaminated pastures is one of the major sources of Pb content in animal tissues [5]. In exposed animals, milk contamination with Pb is caused by the excretion of the metal into milk [5,6,7,8,9]. Numerous authors have reported milk contamination with Pb [6,8,10,11,12,13,14,15].

Primarily, the food chain is subject to Pb contamination through plant uptake from a contaminated environment [16], where animals accumulate Pb in their tissues and excrete it in milk [17]. Because cow milk is considered a good biological matrix, it can thus be used to evaluate the Pb exposure risks in both animals and humans [18]. On the other hand, metal residues in milk could be an important “direct indicator” or an “indirect indicator” of the hygienic status of the milk and the degree of environmental pollution [11]. Apart from environmental contamination and seasonal variations, climatic conditions, lactation period of cows, health conditions, and animal annual feed composition have been identified to have a significant effect on the variability of the mineral content in raw cow milk [11].

Milk is regarded as nature’s single most complete food [19]. It is the most valuable and regularly consumed food that is best known for its good composition of major minerals, proteins, fats, and vitamins [20]. As a basic food in the human diet, it is consumed both in its unprocessed form and as various dairy products [11]. Since milk is largely consumed by infants and children, Pb residues in milk are of particular concern [21], especially in children where even low-level exposure in early childhood is known to cause cognitive development deficits [22]. In adults, Pb poisoning causes multiple health effects, including anemia, hypertension, renal damage, immunotoxicity, etc. [3]. Chronic exposure to Pb is also associated with spontaneous abortions and stillbirth in women [13], as well as infertility in men [23]. 

High levels of environmental Pb are attributed to anthropogenic activities such as mining and smelting, battery manufacturing, recycling of waste batteries, and burning coal [24]. In Kabwe, the capital of Zambia’s Central Province, extensive Pb pollution in the nearby townships caused by emissions from a Pb–Zn mine and smelters has been reported [25,26]. As a result, high Pb concentrations have been detected in animals, including livestock and dogs [26,27,28,29,30,31,32]. In addition, high levels of Pb concentrations in humans, especially children, have been reported [33]. Despite these numerous reports about Pb contamination in Kabwe, dietary Pb exposure and its associated risks have not been evaluated. To address this knowledge gap, the current study was conducted to: (1) quantify the Pb concentrations in milk from cattle reared around the Pb–Zn mine in Kabwe; (2) investigate the seasonal variations of Pb concentrations in cow milk; (3) assess the health risk impact of exposure to Pb through cow milk consumption by the local residents of Kabwe.

## 2. Methods and Materials

### 2.1. Location of the Study Area and Study Design

The current study was conducted using a convenient sampling technique. The farms located in the proximities of the mining area in Kabwe town in Central Province were selected as study sites; Chongwe town in Lusaka Province Zambia was included as a reference site (a non-mining area). The geographical location of Kabwe is 14°27′ S and 28°26′ E, about 150 km north of Lusaka, the capital city of Zambia; Chongwe is located at 14°27′ S and 28°26′ E, 45 km east of Lusaka.

The focus of the study was on five regions of Kabwe mainly with a target selection of farms composed of traditional smallholder milk producers, emerging dairy farms of small and medium-size, namely Kang’omba (KN), Kafulamse (KF), Mpima (MP), Mukobeko (MK), and Munga (MN) (Figure 1). Chongwe (CN) town in the Kanakantapa area, as it is a non-mining area, was considered a reference zone for comparative purposes with the samples from Kabwe. The sampling locations were marked using the Global Positioning System (GPS) (Appendix A). The sampling zones were chosen based on the high dependence on traditional or mix-breed cattle reared using free-range practices by farmers for both beef and milk production. According to Mumba et al. [34], Kabwe produced about 1,610,700 L of milk per year. Farms having a herd size greater than or equal to five lactating cows were registered for sampling. If the farm had more than five but less than ten lactating cows (5 ≥ 10), all the available lactating cows were considered as study subjects, as long as they met the selection criteria (Appendix A); however, in a case where lactating cows exceed 10, the extra number of subjects was calculated based on the 10 percent fraction of the total herd to cow ratio, following random selection criteria.

### 2.2. Milk Sampling

Cow milk sampling was conducted between 2018 and 2021 during the wet and dry seasons. Generally, seasons are divided into two distinctive halves, a dry half from May to October and a wet half from November to April. The coldest month is July, with temperatures in the range of 3.6–12.0 °C, while the hottest month is October, with temperatures averaging 27.7–36.5 °C. On average, annual rainfall ranges from 700 mm in the extreme southwest to 1400 mm in the north and is 1001 mm on average [35]. To evaluate seasonal variations of Pb in the milk of the study participants, samples were collected in a follow-up design study in February/March (wet season) and October (dry season). Samples were collected during morning routine milking [36] from healthy cows according to the standard methods [37,38]. Before sample collection, the udder of each cow was washed using distilled water and about 10 mL was expressed manually directly into sterile 15 mL polypropylene tubes (SuperClear^R^ Labcon, Petaluma, CA, USA). After collection, the samples were then homogenized by inverting the tubes ten times and temporarily stored in a cooler box with ice packs before transportation for storage at −20 °C at the University of Zambia, School of Veterinary Medicine, Lusaka, Zambia until analysis. 

### 2.3. Sample Preparation and Microwave Acid Digestion

For the preparation of all solutions, double distilled water from a Milli-Q-Element system (18 MΩ·cm, Millipore^®^, Milford, MA, USA) was used. Metal-free polypropylene vials (Kanto Chemical Co., Inc., Tokyo, Japan) were pre-cleaned with 2% diluted HNO_3_ (Kanto Chemical Co., Inc., Tokyo, Japan) for 24 h and rinsed thoroughly with Milli-Q double distilled water before use. The milk samples were thawed at room temperature and homogenized by vortex. Approximately 1 g of milk was accurately weighed in a pre-cleaned Berghof digestion vessel (DAP-60K, Enigen, German), followed by the metal extraction in a closed microwave digestion system (Berghof, Speed Wave® ENTRY, Eningen, Germany) as described by Toyomaki et al. [27]. Briefly, metal extraction was conducted using a closed, optimized microwave digestion system under automated temperature and pressure-control conditions for 31 min (Appendix A) after the addition of 5 mL of 30% nitric acid (69% HNO_3_ pure, HI-AR^TM^, Kanto Chemical Co., Inc., Tokyo, Japan), and 1 mL of 30% hydrogen peroxide (Kanto Chemical Co, Inc., Tokyo, Japan). The digested samples were then cooled for 20 min and transferred into 15 mL sterile polypropylene tubes. To ensure homogeneity, the contents were thoroughly mixed by inverting the tube for ten min. Similarly, blank samples were also prepared alongside the milk samples to maintain the uniformity of digestion parameters.

### 2.4. Lead Determination in Milk

Lead analysis in milk was performed using Zeeman-corrected graphite furnace absorption spectrophotometry (GFAAS), (Hitachi-Zeeman 2010 model, High Technologies Corporation, Tokyo, Japan) equipped with a graphite furnace according to a technique described by Yabe et al. [26]. In this graphite method of Pb analysis, a matrix modifier of 0.5% ammonium hydrogen sulfate (NH_4_H_2_PO_4_) (Nakagyo-Ku, Kyoto, Japan) was used to minimize the matrix effect according to the manufacturer’s guide. Concomitantly, the auto-programmed sampler per time injected 20 µL of sample and the modifier for analysis. The operation conditions of the GFAAS are as given in Appendix A.

### 2.5. Quality Assurance

The Pb analysis by the GFAAS method included linear range, coefficient of correlation limits of detection (LOD), and limits of quantification (LOQ) analytical parameters. The LOD value was calculated by multiplying 3 times standard deviation (SD) for 10 replications of the blank, while the LOQ was calculated by multiplying 10 times SD of the slope/intercept (Appendix A). Replication of blank samples analysis was measured directly, and their measurements were subtracted from the sample intensities [39]. Method validation and accuracy were performed according to Yabe et al. [26] using DOLT-5 (Dogfish liver, National Research Council of Canada, Ottawa, ON, Canada). Replicate analysis of DOLT-5 reference material indicated the accuracy and recovery rate of 93.1–119.8% (relative standard deviation, RSD ≤ 5%). The confidence in the integrity of data was increased by duplicate measurement of samples in µg/kg (ppb) anchored on a multipoint calibration curve prepared from 1000 mg/L of Pb (Himea, New Deli, India) stock solution.

### 2.6. Data Analysis

Descriptive statistical analysis of the data was performed using GraphPad Prism software (Prism 7 for Windows; Version 5.02, GraphPad Software, Inc., San Diego, CA, USA). The explorative parameters are presented as n—sample number; m—mean value; (SD±)—standard deviation; Min-Max—range of Pb levels in milk. Prior to analyses, data were examined for normality of distribution by Kolmogorov–Smirnov normality test. The test performed showed the departure of data from normality. To stabilize the variances, we transformed the data by a base 10 logarithm. Statistical comparison analysis was performed to determine the differences in the mean concentrations of Pb in the milk samples collected in the wet and dry seasons and among different sampled regions using analysis of variance (ANOVA) and Tukey’s multiple comparison test, respectively. All significant differences were set at *p* < 0.05.

### 2.7. Probabilistic Health Risk Assessment of Lead in Milk

To evaluate the Pb exposure through ingestion of contaminated cow milk in the local residents of Kabwe, the risk assessment was performed based on chronic daily intake (CDIs), target hazard quotients (THQs), and incremental lifetime cancer risk (ILCR) parameters according to the guidelines prescribed by the United States Environmental Protection Agency (USEPA) [40].

#### 2.7.1. Chronic Daily Intake

The CDI is a value related to the metal concentration in milk, the daily consumption of milk, and the body weight of the consumer, which influences tolerance to a contaminant [1]; in this case, Pb. The CDI values were determined according to Equation (1) [41].
CDI = (C*_m_* × D)/BW(1)
where CDI is the estimated chronic daily intake of Pb (µg/kg ^−1^ BW day^−1^); C*_m_*, mean Pb concentration (µg/kg); D, daily milk intake (µg/day); and BW is the average body weight (kg). Owing to the scarcity of updated data on the consumption rates of milk in Zambia, the average daily intake of milk in children and adults is 21 and 17 g/day/person for body weights of 10 kg and 70 kg, respectively [41,42], were used for the calculations in the present study.

#### 2.7.2. Risk Characterization

The risk characterization analysis using carcinogenic and non-carcinogenic risk assessment via ingestion was considered an important tool for identifying the health risk effects in humans and providing risk evidence for decision-making.

##### Non-Carcinogenic Risk

Non-carcinogenic risk is evaluated by comparing an exposure level (dose) over a specified period, such as a lifetime, with a reference dose derived for a similar exposure period [43].

The non-carcinogenic risk is characterized in terms of THQ, which has been recognized as a useful parameter for the evaluation of risks associated with the consumption of metal-contaminated food. Thus, the potential non-carcinogenic risks for the exposure to Pb via consumption of cow milk are assessed by comparison of CDI from the oral exposure route with the chronic dose (RfD) to find the THQ value described by the USEPA [40] as follows:THQ = (CDI/RfD) × 10^−3^(2)
where THQ is unitless, CDI is the chronic daily intake average (mg/kg/day); and RfD is mg/kg/day. The oral reference dose for Pb is 3.5 × 10^−3^. THQ > 1 assumes that there may be a concern for potential non-carcinogenic risks. In a case where THQ < 1, it means that the hazard is unlikely to cause adverse health effects for the exposed populations [43].

##### Carcinogenic Risk

Carcinogenic risk refers to the incremental probability of an individual developing any kind of cancer in a lifetime because of exposure to carcinogens [43]. In the current study, carcinogenic risks were calculated as the incremental probability of an individual developing cancer over a lifetime as a result of exposure to Pb according to the linear equation [36,43,44] as follows:Carcinogenic risk = CDI × CSF(3)
where CDI is the daily chronic intake; CSF, cancer slope factor (mg/kg/day)^−1^. According to the USEPA [40], the acceptable safe risk ranges from 10^−6^ to 10^−4^.

## 3. Results and Discussion

### 3.1. Lead Concentrations in Cow Milk

The overall summarized concentrations of Pb analyzed in cow milk samples from the studied regions are presented in Table 1. The mean Pb levels detected in cow milk varied in the range of 0.60–2.32 µg/kg in the wet season and 0.50–4.24 µg/kg in the dry season. The lowest mean value recorded was in Mukobeko, which surprisingly showed a slightly lower mean value than the level that was recorded in Chongwe, the reference site. According to the results, Munga had the highest mean Pb concentration. Table 1 below shows the descriptive results obtained in the study.

The variation of mean Pb concentration results in cow milk obtained showed significant differences among the sampled regions when compared with Tukey’s multiple comparison test set at 0.05 (*p* < 0.05). Data in Figure 2 show the graphical variations of mean Pb concentrations obtained in each sampled region.

The general trend of the mean Pb concentrations in milk during the wet season followed the decreasing order of Kang’omba > Munga > Mpima > Mukobeko > Kafulamase > Chongwe, while in the dry season, the trend followed the order of Munga > Kang’omba > Kafulamase > Mpima > Chongwe > Mukobeko as illustrated in Figure 3. The mean Pb concentrations measured in milk samples in the dry season were higher than the samples in the wet season, especially in farms closer to the mine.

Based on the results of the current study, Pb was present in all sampled sites, including Chongwe (reference site). The source of Pb in cattle from the sampled farms around Chongwe town, which has no history of Pb mining, is unknown; however, the current study indicates that Chongwe, the reference site, had the least Pb results, and correspondingly, the interval Pb concentration was remarkably minimal (Figure 3). The milk Pb concentrations in the mining areas of Kabwe showed higher variability compared to the non-mining in Chongwe (Figure 3).

Generally, the findings indicate that the spatial distribution pattern of the Pb contamination in the cow milk samples investigated corresponded with the prevailing wind direction and the distance to the point source of the Pb pollution. For example, Munga on the west and Kang’omba on the southwest were subject to high Pb pollution because of their proximity and geographical location to the Pb–Zn mine. The most likely explanation for the elevated Pb concentrations in the two areas compared to other sites in the dry season could be due to the airborne Pb in dust and contaminated forage produced on agricultural surfaces or cattle grazing on pastures contaminated by emissions from smelters [5].

On the other hand, the least Pb concentration in the current study was observed in the dry season in Mukobeko (0.5 µg/kg), followed by Mpima (0.84 µg/kg), regions that were located on the northern and northwestern sides of Kabwe town. Interestingly, the highest Pb concentration (10.8 µg/kg) during the same season was recorded in Kafulamase, a sampling location that was further on the southeastern side of the Pb–Zn mine. Studies reveal that large amounts of tailings and wastewater produced in the mining process account for high metal contamination in the surrounding environment [43]. Mostly, near the smelters [45], there is an increased uptake of Pb through contaminated fodder, which subsequently results in enhanced excretion of Pb residues in milk [12]. Moreover, studies indicate that rainwater aids metal dispersion, causing widespread contamination and affecting agricultural fields and water bodies [2]. Thus, intake of Pb contaminated water emanating from the Pb–Zn mine carried downstream during the rainy season could be associated with the appreciably high Pb content in milk observed from the Kafulamse region, 15 km away from the mine. The overall results in the present investigation showed that Pb levels in cow milk were lower than most of the findings that have been reported previously from various countries (Table 2).

Compared to the established values in the present study, other authors in Romania [1], India [55], Iran [49], and Kazakhstan [46] recently reported higher Pb concentrations of 24.0 ± 15.0 µg/kg, 209 ± 0.70 µg/kg, 32.83 ± 20.80 µg/kg, and 11.6 ± 10 µg/kg, respectively. Nonetheless, values obtained in the current study were remarkably higher than those reported by Elsaim and Ali [50] in Sudan but within the range that was found in China (3.60 ± 2.30 µg/kg) by Wang et al. [24]. Although Pb was detected in all Kabwe regions in our present study, the concentrations found in cow milk were below the 20.0 µg/kg safety standard threshold set by Codex Alimentarius Commission [56], and 100 µg/kg, the benchmark value established by the European Food Safety Authority of the European Union (EU) [57].

### 3.2. Seasonal Variations of Lead Concentration Trends in Milk

The general trends of Pb concentrations observed in each region are presented in Figure 4a–d below.

The study showed varied Pb concentration trends in each season, despite the animals generally sharing the same grazing pasture in each region. For example, in Kan’gomba and Munga, the paired results indicate that the mean Pb concentrations in the dry season were higher than in the wet season. Similarly, the results obtained in Kafulamase in the paired samples analyzed indicate remarkable high Pb content in the milk in the dry season compared to the results obtained in the wet season; however, the mean Pb concentration trends in Mukobeko and Mpima were similar but different from the pattern observed in Kang’omba, Kafulamse, and Munga. On the contrary, in Chongwe (reference area), samples analyzed showed an irregular Pb pattern, although the concentrations seemed to oscillate within the same magnitude in each season.

The difference in seasons evoked considerable Pb concentration variations either between the region in the same sampling sites and/or among different regions. Factors such as the location and distance of the farms from the Pb–Zn mine could have also contributed to the observed Pb concentration fluctuations. The literature indicates that the degree of heavy metal contamination in cow milk is not constant but differs depending on the exposure routes, environmental condition, animal’s nutrition, stage of lactation, and animal breed [58]. Since the present study focused on cattle with mixed breeds reared on free-range practices, variability of Pb residues in milk in different regions was expected.

### 3.3. Health Risk Assessment

Health risk is defined as the likelihood of harmful effects on human health as a result of environmental pollution [43]. Thus, exposure assessment is the process of measuring or estimating the intensity, frequency, and duration of human exposures to an environmental agent [59]. Exposure assessment results can be used to evaluate risks to human health [24]. Resulting from such an evaluation process in the current study, the summarized data are presented in Table 3, Table 4 and Table 5 below.

#### 3.3.1. Chronic Daily Intake of Lead Metal in Milk

Table 3 below shows the summary of the CDI values calculated based on the average concentrations found in cow milk analyzed from each sampled region. These were compared with provisional tolerable daily intake as set by the Joint FAO/WHO Expert Committee on Food Additives (JECFA) [60]. The results obtained in the present study showed comparatively high CDI for children than for adults; however, in both cases, the values were all within the permissible limits. The CDI results established in the present study ranged from 5.10 × 10^−7^ to 4.98 × 10^−6^ in children, while in adults, it ranged from 1.75 × 10^−7^ to 1.487 × 10^−6^; however, the highest CDI value (4.98 × 10^−6^) was recorded for children in the milk samples that were collected in the Kang’omba region during the dry season, whereas the lowest value (1.75 × 10^−7^) in adults was obtained in samples that were collected from the Mukobeko region in the same period (Table 3).

The average CDI results in children both in the wet season and dry indicate higher values than in adults (8.75 × 10^−6^ > 2.71 × 10^−6^ and 1.19 × 10^−5^ > 3.74 × 10^−6^) for the same period, respectively. In all cases, CDI values were below the oral reference dose of 3.57 µg/kg/day of Pb [21]. According to the present results, the observed CDI trends regionally in both children and adults were found in the decreasing order of Kang’omba > Munga > Kafulamase > Mpima > Chongwe > Mukobeko > in the dry season.

The CDI values established in the present study were lower than the values found in Africa by Salah et al. [61] and Meshref et al. [62] of 158.5 and 1.70 × 10^−4^ µg/kg/day, respectively. Similarly, other authors [38,63] in Europe reported a CDI range of 4.40 × 10^—4^ to 8.00 × 10^−5^, which was several folds higher than what was found in the current study. On the contrary, CDI values of 5.40 × 10^−6^ and 3.40 × 10^−5^ that were similar to the current study were reported in Asia by Muhib et al. [20] and Norouzirad et al. [51], respectively. On the other hand, Ismail et al. [53] reported much higher values in the range of 0.069 to 0.946 in both children and adult milk consumers.

Food intake has been identified to be the major route of human exposure to Pb, although, for children, ingestion of soil and dust can also be an important contributor [56]. In other studies, it is reported to account for over 90% compared to other exposure pathways such as inhalation and dermal contact [64]. Consequently, in a general population, the most probable route of exposure that leads to elevated blood Pb levels is ingestion [3]. Children are at a greater risk of chronic Pb exposure and poisoning than adults due to their higher CDI coupled with lower body weight [13,65]. There is no established tolerable Pb intake for which Pb cannot cause adverse effects [60]; however, the U.S. Food and Drug Administration (FDA) recommends interim reference levels of 3 and 12.5 µg/day for children and adults (women of childbearing age), respectively. This corresponds to the blood lead level (BLL) of 0.5 µg/dL for a general population [66]. Analysis of samples in the present study, however, showed that the calculated values of Pb in cow milk were lower than the interim reference benchmarks recommended by FDA.

Although the low levels were found to be below the recommended risk level by the FDA in cow milk from Kabwe, potential risks may probably occur due to dietary and non-dietary contributing factors that may magnify the Pb exposure effects in the population; therefore, strict regular monitoring of Pb contamination of milk and milk products is recommended [61] for the food safety and quality aspects of the consumers.

#### 3.3.2. Evaluation of Non-Cancer Risk Assessment for the Kabwe Population

The calculated potential adverse effects for the current study associated with non-carcinogen Pb in cow milk are presented in Table 4. The estimation of the maximum permissible risk on the human population was calculated based on THQ according to Kasozi et al. [44].

Target THQ is a method developed by the US environmental protection agency (EPA) for estimation of potential human health risk exposure to chemical pollutants associated with non-carcinogens [58], as expressed in Equation (2). The average THQ values in children and adults both in the wet season and dry were higher than in adults (7.75 × 10^−4^ to 2.5 × 10^−3^) and 1.07 × 10^−3^ to 3.41 × 10^−3^), respectively. As shown in Table 4, 1.42 × 10^−3^ was the highest THQ value calculated in children for the milk samples collected in the Kang’omba region in the dry season, whereas the highest value of 4.24 × 10^−5^ in adults was calculated in the analyzed samples from the Munga region in the same season. Considering the THQ value of less than one as non-hazardous to the consumers [36], and since all the THQ values obtained in the current study were consistent with this benchmark, consumption of cow milk in the sampled regions was considered safe from non-cancer risk effects due to Pb exposure. Our findings with a similar matrix agree with the results reported in Asia by Abedi et al. [65] in Iran (0.009 to 0.032) and those found in South America by Gonzalez et al. [36] in Mexico (0.039 to 0.059).

Similarly, results in the present study were consistent with the findings reported in Asia (1.2 × 10^−3^ to 9.8 × 10^−3^) [20,51]; however, in Europe, the THQ values reported ranged from 1.3 × 10^−2^ to 7.6 × 10^−2^ [38,63,67]. Although these values from the reported regions were also lower than the safe limit of THQ > 1, they were slightly higher than the findings in the current study. In Uganda, Kasozi et al. [44] also found far much higher values in children and adults, ranging from 6.2648 to 2.116, respectively, compared to our results.

#### 3.3.3. Evaluation of Carcinogenic Risk Assessment for the Kabwe Population

The cancer risk assessment was evaluated as ILCR, and the results are indicated in Table 5.

The ILCRs caused by the ingestion of Pb in cow milk obtained in the present study for both children and adults on average ranged from 7.43 × 10^−8^ to 1.01 × 10^−7^ and 2.31 × 10^−8^ to 3.18 × 10^−8^, respectively. Our ILCR results obtained in the study indicate that they were far lower than the results reported by Kasozi et al. [44] in Uganda as 0.847 × 10^−4^ and 0.283 × 10^−4^ in children and adults, respectively. Further, Abedi et al. [65] in Iran reported higher values of ILCR than findings in the current study in children and adults, ranging from 1.89 × 10^−3^ to 2.45 × 10^−3^ and 2.96 × 10^−4^ to 3.85 × 10^−4^, respectively.

For safety reasons, incremental lifetime cancer risks ranging from 1.0 × 10^−6^ to 1.0 × 10^−4^ are acceptable [68]. In the current study, both in the wet and dry seasons, the ILCR results were negligible and far lower than the reference range given. The ILCR values, therefore, suggested that cow milk from Kabwe did not pose carcinogenic risks to human consumers; however, long-term exposure to metals such as Pb and their complete absorption through the digestive tract is considered to be the worst-case scenario [1] because of their cumulative deleterious effect on human health [69]; therefore, regular assessment of their dietary exposure levels and impact to humans is essential.

Although the current study results indicated that consumption of cow milk in smallholder farming communities did not constitute a health hazard to consumers according to the Codex Alimentarius Commission set standard [56], the study had limitations. Limitations included a small sample size in the Munga region due to a small animal population, which could have some effect on the statistical analysis of data. Moreover, a shorter lactation period among traditional cow breeds compared to dairy cows, as well as the outbreak of foot and mouth disease during the sampling period, made the follow-up sampling study design a great challenge.

## 4. Conclusions

The intake of cow milk from free-range cattle in the studied regions indicated that the adverse health effects were not likely to be significant, based on the CDI, THQ, and ILCR parameters measured due to low Pb exposure in milk. The current study also revealed that the change of season did not cause a significant difference in mean Pb concentrations in Kabwe; however, since Pb is accumulative and can cause adverse health effects to humans even in low-dose exposure, regular assessment is necessary for quantifying the risks. Considering also the important influence of the aggregate potential risk factors due to dietary Pb exposure, a broad-based questionnaire survey is recommended to determine the consumption levels of cow milk in the studied regions. Furthermore, evaluation of a characteristic wide range of toxic heavy metals should also be considered for the determination of accurate combined toxic metal exposure effects that may be associated with the consumption of cow milk, including other dietary sources such as vegetable crops grown in the affected areas.

Further, given the high variability of results in certain sampled sites, even in places far away from the Kabwe Pb–Zn mine, further seasonal evaluation of Pb concentrations in animal drinking water from various sources such as ponds/dams, wells, streams, and rivers, including soil and fodder from the individual farms, is strongly recommended.

## Figures and Tables

**Figure 1 ijerph-19-04757-f001:**
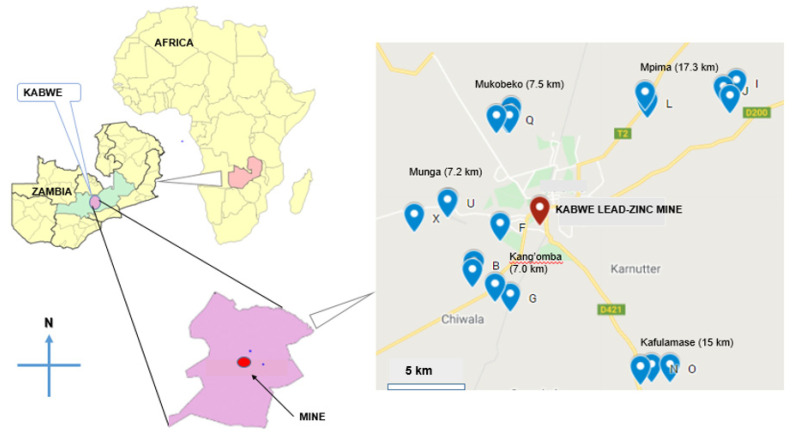
GPS map (modified from Google My Maps) showing sampling locations around the Pb–Zn mine in five regions of Kabwe (Kang’omba n = 56; Mpima, n = 54; Kafulamase, n = 35; Mukobeko, n = 59; Munga, n = 10).

**Figure 2 ijerph-19-04757-f002:**
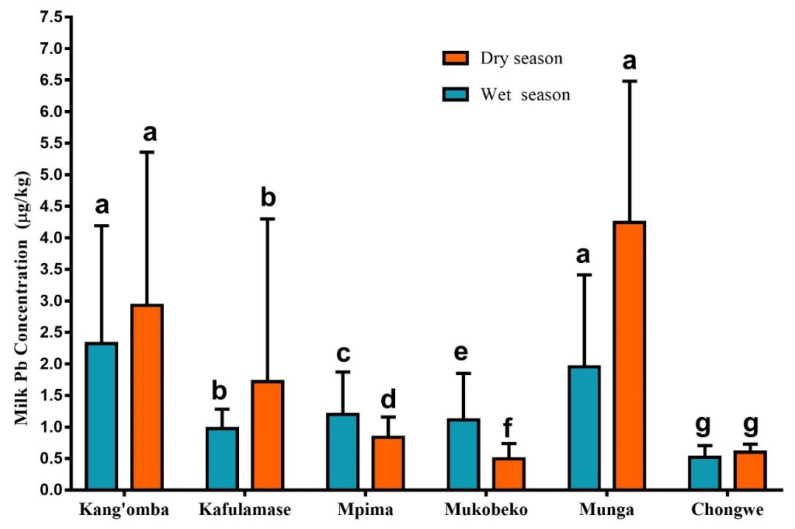
The variation of mean Pb concentration results in cow milk (µg/kg-wt./wt.) per region and per season. The lower-case letters a–g represent significant differences among the sampled regions using Tukey’s multiple comparison test set at 0.05 (*p* < 0.05).

**Figure 3 ijerph-19-04757-f003:**
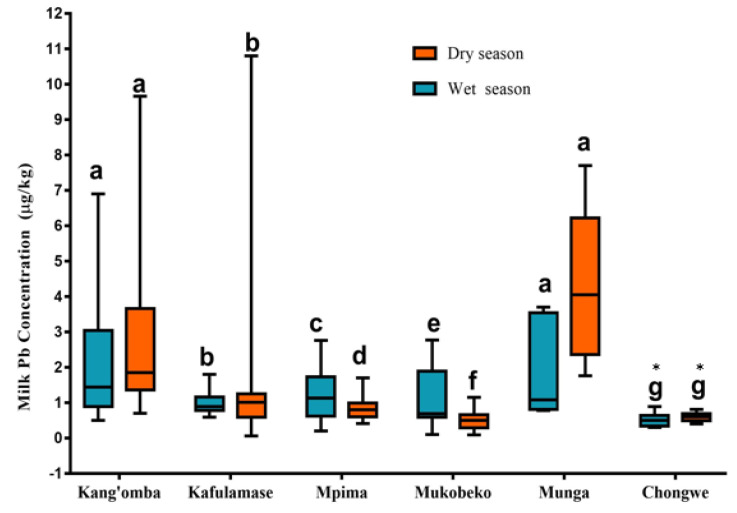
A summary of Pb concentrations (µg/kg) in cow milk analyzed from different sites per season in the mining area (Kang’omba, n = 56; Mpima, n = 54; Kafulamase, n = 35; Mukobeko, n = 59; Munga = 10) and in a non-mining area of Chongwe (reference site), n = 18. Data are presented in box and whisker plots: lines marked across the boxes indicate the medians; box limits represent 25th and 75th percentiles; ends of the whisker show minimum and maximum values of Pb concentrations measured. The lower-case letters a–g represent significant differences among the sampled regions using Tukey’s multiple comparison test (*p* < 0.05). Asterisk (*), means reference site.

**Figure 4 ijerph-19-04757-f004:**
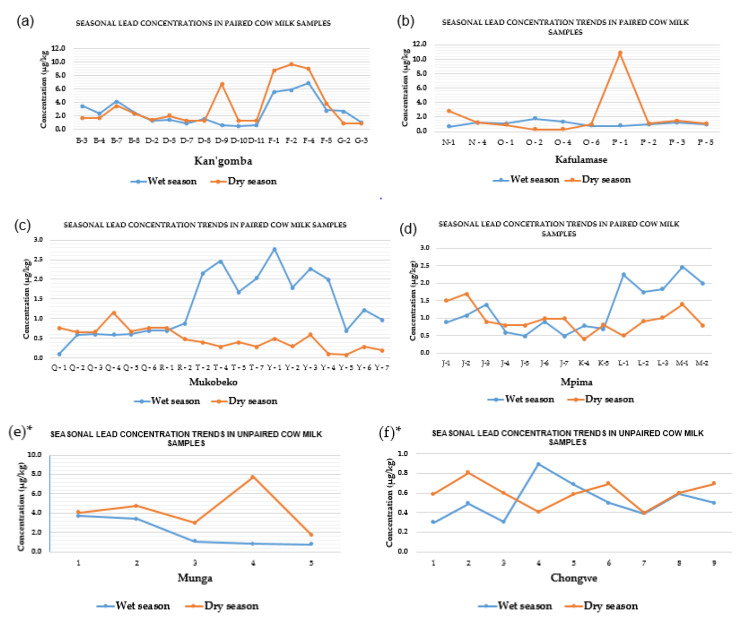
Pb concentration trends in paired cow milk samples collected in the wet and dry seasons in: (**a**) Kang’omba; (**b**) Kafulamase; (**c**) Mukobeko; (**d**) Mpima regions. Asterisk on (**e**)* and (**f**)* indicates unpaired milk samples, i.e., milk samples collected from different selected subjects in Munga and Chongwe regions during the wet and dry seasons. The letters on graph (**a**–**d**) represent identity numbers of paired samples from the same individual cows, while the numbers on graph (**e**)* and (**f**)* represent sample identity numbers for unpaired samples from different animals of the same herd.

**Table 1 ijerph-19-04757-t001:** Summary of mean lead concentrations (µg/kg-wt./wt.).

Season	Region	No. of Farms	Samples (n)	Mean (m)	Standard Deviation (SD±)	Min-Max (µg/kg)
	Kan’gomba	5	22	2.32	1.87	0.05–6.90
**Wet**	Kafulamase	3	16	0.98	0.30	0.59–1.80
**season**	Mpima	5	28	1.20	0.67	0.20–2.76
	Mukobeko	5	29	1.11	0.74	0.10–2.77
	Munga	3	5	1.96	1.46	0.78–3.70
	Chongwe	3	9	0.60	0.19	0.89–4.66
	**Average**			**1.36**	**0.88**	**0.05–6.90**
	Kan’gomba	4	34	2.93	2.43	0.70–9.66
	Kafulamase	3	19	1.72	2.58	0.06–10.8
**Dry**	Mpima	5	26	0.84	0.32	0.41–1.70
**season**	Mukobeko	5	30	0.50	0.24	0.09–1.15
	Munga	3	5	4.24	2.24	1.76–7.70
	Chongwe	3	9	0.51	0.13	0.40–0.81
	**Average**			**1.79**	**1.32**	**0.06–10.8**

**Table 2 ijerph-19-04757-t002:** Lead levels in cow milk from various countries in recent five years (2015–2021).

Year	Country	Source of Milk	N	Method Used	Mean (µg/kg)	Range	Reference
2021	Zambia	Farms near the mining area	233	GFAAS	2.22 ± 1.89	LOD-9.66	Present study
2020	Kazakhstan	Farms	120	AAS	11.6 ± 10.0	-	[46]
2020	Slovakia	Farms	40	ETA-AAS	10.0	-	[47]
2019	China	Wholesale markets and stores	208	ICP-MS	3.6 ± 2.30	-	[48]
2019	Romania	Small cattle farms	10	GFAAS	24. ± 15.0	0.010–0.048	[1]
2018	Iran	Dairies and markets	36	ICP-OES	32.83 ± 20.80	15.70–68.0	[49]
2018	Sudan	Farms	9	AAS	<LOD	-	[50]
2018	Iran	Dairy, industrial and traditional farms	118	GFAAS	47.0 ± 3.9	N.D-250	[51]
2018	Uganda	District	20	AAS	10.48 ± 1.82	6.62–14.34	[44]
2017	India	Farms	30	ICP-AES	124.0	0.016–0.356	[52]
2017	Pakistan	Farms	240	AAS	0.021	0.007–0.041	[53]
2017	Egypt	Dairy shops	18	AAS	93.4 ± 18.8	0.007–0.341	[54]
2017	Mexico	Sub-basin of industrial and urban region	40	ICP-OES	46.0± 28.	0.039–0.05	[36]
2016	Bangladesh	Branded, dairy farm and household farmers	27	FAAS	0.012 ± 0.001	-	[20]
2016	Romania	Rural area	19	ICP-MS	15.8 ± 5.45	0.01–0.48	[38]
2015	Egypt	Dairy shops and groceries	30	GFAAS	414.0 ± 34.2	0.09–0.52	[15]

**Table 3 ijerph-19-04757-t003:** Chronic daily intake (CDI) of lead through consumption of milk for the resident children and adults of Kabwe.

	Wet Season CDI (µg/kg/day)		Dray Season CDI (µg/kg/day)	
**Region**	**Children**	**Adult**	**Children**	**Adult**
Kan’gomba	3.77 × 10^−6^	6.66 × 10^−7^	4.98 × 10^−6^	8.79 × 10^−7^
Kafulamase	8.33 × 10^−7^	3.43 × 10^−7^	1.46 × 10^−6^	6.02 × 10^−7^
Mpima	1.02 × 10^−6^	4.20 × 10^−7^	1.02 × 10^−6^	4.20 × 10^−7^
Mukobeko	9.44 × 10^−7^	3.89 × 10^−7^	4.25 × 10^−7^	1.75 × 10^−7^
Munga	1.67 × 10^−6^	6.86 × 10^−7^	3.60 × 10^−6^	1.48 × 10^−6^
Chongwe *	5.10 × 10^−7^	2.10 × 10^−7^	4.34 × 10^−7^	1.79 × 10^−7^
**Average**	**8.75 × 10^−6^**	**2.71 × 10^−6^**	**1.19 × 10^−5^**	**3.74 × 10^−6^**

* indicate reference site.

**Table 4 ijerph-19-04757-t004:** Target hazard quotient (THQ) of lead through consumption of cow milk (in children and adults) in the residents of Kabwe.

	Wet Season THQ		Dry Season THQ	
**Region**	**Children**	**Adult**	**Children**	**Adult**
Kan’gomba	1.08 × 10^−3^	1.90 × 10^−4^	1.42 × 10^−3^	2.51 × 10^−4^
Kafulamase	2.38 × 10^−4^	9.80 × 10^−5^	4.18 × 10^−4^	1.72 × 10^−4^
Mpima	2.91 × 10^−4^	1.20 × 10^−4^	2.91 × 10^−4^	1.20 × 10^−4^
Mukobeko	2.70 × 10^−4^	1.11 × 10^−4^	1.21 × 10^−4^	5.00 × 10^−5^
Munga	4.76 × 10^−4^	1.96 × 10^−4^	1.03 × 10^−3^	4.24 × 10^−4^
Chongwe *	1.46 × 10^−4^	6.00 × 10^−5^	1.24 × 10^−4^	5.10 × 10^−5^
**Average**	**2.50 × 10^−3^**	**7.75 × 10^−4^**	**3.41 × 10^−3^**	**1.07 × 10^−3^**

* indicate reference site.

**Table 5 ijerph-19-04757-t005:** Incremental lifetime cancer risk (ILCR) of lead through consumption of cow milk (in children and adults) in the residents of Kabwe.

	Wet Season ILCR		Dry Season ILCR	
**Region**	**Children**	**Adult**	**Children**	**Adult**
Kan’gomba	3.21 × 10^−8^	5.66 × 10^−9^	4.23 × 10^−8^	7.47 × 10^−9^
Kafulamase	7.08 × 10^−9^	2.92 × 10^−9^	1.24 × 10^−8^	5.12 × 10^−9^
Mpima	8.67 × 10^−9^	3.57 × 10^−9^	8.67 × 10^−9^	3.57 × 10^−9^
Mukobeko	8.02 × 10^−9^	3.30 × 10^−9^	3.61 × 10^−9^	1.49 × 10^−9^
Munga	1.42 × 10^−8^	5.83 × 10^−9^	3.06 × 10^−8^	1.26 × 10^−8^
Chongwe	4.34 × 10^−9^	1.79 × 10^−9^	3.68 × 10^−9^	1.52 × 10^−9^
**Average**	**7.43 × 10^−8^**	**2.31 × 10^−8^**	**1.01 × 10^−7^**	**3.18 × 10^−8^**

## Data Availability

Data are available on request to the authors.

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
