# Peer review of "Human Health Risk Assessment from Lead Exposure through Consumption of Raw Cow Milk from Free-Range Cattle Reared in the Vicinity of a Lead–Zinc Mine in Kabwe"

_ijerph, 2022, doi:10.3390/ijerph19084757_

Round 1

Reviewer 1 Report

Title: not very descriptive.  It sounds like a survey of people consuming the milk rather than an analysis of milk contamination.

Introduction:

P2, L53-55:  First sentence “milk is regarded as nature’s single most complete food” should be referenced.  The sentence that follows about milk composition is referenced (19) to a paper about heavy metal contamination of milk, so maybe there’s a better reference for this?

P2, L55:  You should probably define what you mean by “raw,” since I think you’re referencing liquid milk versus other dairy products.  Raw milk can also mean unpasteurized.  This is relatively minor, since pasteurization is unlikely to affect the Pb concentration (though the processes of making yogurt, cheese, etc will likely increase Pb in the final product).

P2, L63-54:  Is there a history of Pb paint in Kabwe?  Most of the Pb exposures in cattle in my part of the world are from old paint or discarded automotive batteries.

P2, L68:  “alarming levels of Pb concentrations” is sort of redundant.

M&M:

P2, L91:  “on” is not the right word for this sentence, maybe “reared using free-range practices” or something.

P2-3, L95-97:  According to this, you sampled every cow in herds up to 10, but only 10% of cows “if the number of lactating cow was high” so does that mean if the number was 11, you only sampled one cow?  Unless the “number of lactating cows was high” means >100, it doesn’t seem like 10% would be a reasonable sample size.

P4, L179:  This uses reference 39, a USEPA risk assessment, but I was unable to find the reference using the information you list for reference 39. 

P5, L189: typographical error: calculations

P5, L194: omit the word “effects”

P5, L205:  I don’t understand the term “non-carcinogenic cancer risk”

Results

P7, L260-263:  I don’t see the word “inevitable” in papers very often.  Maybe star the sentence more like “the most likely explanation of this pattern of Pb distribution is because airborne Pb in dust contaminated forage . . . “

Table 2:  You don’t give the units for this table, and it is a little confusing because different units are used by different studies.

P8, L281-290:  You give your results in micrograms/kg and then give the comparison values in mg/kg or mg/mL.  Can you convert all of the values from the different studies and from the codex and EFSA to the same units so I can understand the comparisons you are making?

Figure 4:  What do the values on the X axis mean?  They are different in the different graphs, so I don’t know if the data can be compared from graph to graph, or only within each graph.

P9, L311:  Typographical error: sampling sites, note sapling sites.

P9, L317:  It wasn’t inevitable, but it was expected.

P10, L332:  The way the numbers are given with dash-lines denotating both the exponent and the separation of values makes them a little more difficult to read.  Can you remove the dash line between values and replace with the word “to”?  Also, can you include the units?

P11, L354-356:  This sentence is misleading.  While I think oral intake is a major route of human exposure, and maybe the most important one, it’s not necessarily associated with food intake.  Lead intake from ingestion of lead dust from paint or other sources and hand-to-mouth contact is a more important cause of exposure than food contamination in some parts of the world.

Tables 4 & 5: Should the last columns be marked “Dry season”?

P12, L407-208 & 411-412:  The formatting for exponential numbers is inconsistent here compared to the rest of the paper.

Conclusions

P12, L428-430:  “not significant” and “insignificant” have slightly different connotations, and you are basing this on very specific parameters.  I think maybe it would be reasonable to say “effects due to exposure to low Pb concentrations in cow milk was not likely to be significant, based on the CDI, THQ, and ILCR values found in this study.” 

P12, L436-439:  I think it would also be prudent to determine if there were other dietary sources of Pb, such as vegetable crops grown in the affected area, or fish from the affected waterways.

References:  I didn’t read these very thoroughly, but there seemed to be a lot of different formatting issues. 

  1. No initials associated with surnames, no period at the end of author list.
  2. No journal name
  3. The year is not bolded here.
  4. I don’t know what the journal’s requirement for using “et al.” is, but I would be surprised if the cut-off is 10 authors.
  5. Capitalization of the manuscript title is different here than in the rest of the document.
  6. Also, here, “cadmium” and “lead” would not be capitalized.
  7. Capitalization of the manuscript title is different here than in the rest of the document.

23: Capitalization of the manuscript title is different here than in the rest of the document.

17:  I don’t understand the convention for abbreviating health.  Here it is “Health” and elsewhere it is “Heal.”  Maybe double- check the proper journal title abbreviations to make sure they are all correct.

28: Capitalization of the manuscript title is different here than in the rest of the document.

32:  Is there supposed to be a “q” after Zambia?

33: I don’t think the journal name should be a url.

36: Capitalization of the manuscript title is different here than in the rest of the document.

37:  I don’t know what kind of reference this is.  There is no journal name or page numbers, and I don’t know what Clujj-Napoca is. 

40: No page numbers given.

43:  No page numbers given.  Also, this and another reference have the year twice and no volume number?

  1. No page numbers.
  2. Capitalization of the manuscript title is different here than in the rest of the document.
  3. No journal
  4. Capitalization of the manuscript title is different here than in the rest of the document.
  5. Capitalization of the manuscript title is different here than in the rest of the document.

54: Capitalization of the manuscript title is different here than in the rest of the document.

56: No punctuation between author and title.  Also, make sure EFSA J is the proper abbreviation.

57: no page numbers.

59:  Spelling error.  Need more information to reference this report correctly.  You should probably include a link to the online version of the document, if possible.

  1. Capitalization of the manuscript title is different here than in the rest of the document.

67:  The journal name is placed before and after the title of the manuscript.

68:  lead and cadmium do not need to be capitalized.

Reviewer 2 Report

This paper seems interesting to me and I have read it with great interest. I am going to make some comments to the authors that I think could improve the paper:

In this work some things have caught my attention, the most striking is the low milk consumption of the inhabitants, 21 g for adults and 17 g for children, it is less than half a glass of milk a day, I thought it was a mistake, but I have gone to the original source [41] and that is indeed the case, here in Spain the consumption is almost 500 g a day, with such low consumption the study makes little sense. Why isn't more milk consumed if there are cows?

An important part of the study focuses on the difference between the dry season and the wet season. What is the working hypothesis regarding seasonality? In the rainy season there should be more lead that moves with the runoff water?

It seems to me very appropriate to have a control area as a background, Chongwe, although it already has certain lead values, higher than those of one of the selected areas Mukubeko, perhaps it is edaphic or geological values, of the original material, is it the same soil or original material as that of the areas studied? It is that if it is not the same type of soil it is not a good control.

Section 2.3 Reagents could be combined with 2.4 and even with 2.5, I believe that in the methodology it is not necessary to specify as much detail as the quality of milliQ water, which is supposed to be present, or of the reagents HNO3 or H2O2 . The three sections can be joined and summarized.

Table S2, A and B are the unfilled templates, why have they not been filled in with the data in the supplementary material? Empty they have no meaning. Attach with the data that has been collected “in situ”.

I suppose that the cows graze all day in the field?

The lead data show a very high deviation.

I find the inclusion of Table 2 very convenient to compare with the values ​​in other countries.

Regarding the bibliographical references, review:

1 Miclean; Cadar… put the acronyms of the names of the authors.

15 El-Ansary, M. instead of El-Ansary, María put the abbreviation.

32 Yabe, J…….instead of putting et al., write the missing names. Cross out Chemosphere, which is repeated.

37 Oana, C…..does not adapt to the format, the name of the paper is in italics and the year is not in its place.

45 Sarsembayeva, N.B…..the name of the journal is missing.

50 Norouzirad, R……..remove et al. and put the name of the missing author.

51 Patel, H.B. ……the name of the journal is missing.
